# Actin Stress Fibers Response and Adaptation under Stretch

**DOI:** 10.3390/ijms23095095

**Published:** 2022-05-03

**Authors:** Roberto Bernal, Milenka Van Hemelryck, Basile Gurchenkov, Damien Cuvelier

**Affiliations:** 1Cellular Mechanics Laboratory, Physics Department, SMAT-C, Universidad de Santiago de Chile, Santiago 9170124, Chile; milenkavh@gmail.com; 2Institut du Cerveau et de la Moelle Épinière, Hôpital Pitié Salpêtrière, 47 bd de l’Hôpital, 75013 Paris, France; basile.gurchenkov@icm-institute.org; 3Sorbonne Université, Faculté des Sciences et Ingénierie, UFR 926 Chemistry, 75005 Paris, France; 4Institut Pierre Gilles de Gennes, Paris Sciences et Lettres Research University, 75005 Paris, France; 5Institut Curie, Paris Sciences et Lettres Research University, Centre National de la Recherche Scientifique, UMR 144, 75248 Paris, France

**Keywords:** actin stress fibers, cell mechanics, SF response

## Abstract

One of the many effects of soft tissues under mechanical solicitation in the cellular damage produced by highly localized strain. Here, we study the response of peripheral stress fibers (SFs) to external stretch in mammalian cells, plated onto deformable micropatterned substrates. A local fluorescence analysis reveals that an adaptation response is observed at the vicinity of the focal adhesion sites (FAs) due to its mechanosensor function. The response depends on the type of mechanical stress, from a Maxwell-type material in compression to a complex scenario in extension, where a mechanotransduction and a self-healing process takes place in order to prevent the induced severing of the SF. A model is proposed to take into account the effect of the applied stretch on the mechanics of the SF, from which relevant parameters of the healing process are obtained. In contrast, the repair of the actin bundle occurs at the weak point of the SF and depends on the amount of applied strain. As a result, the SFs display strain-softening features due to the incorporation of new actin material into the bundle. In contrast, the response under compression shows a reorganization with a constant actin material suggesting a gliding process of the SFs by the myosin II motors.

## 1. Introduction

It is well established that cells are sensitive to their mechanical environment, modifying their activity and shape in response to it. The major actor in this process is the actomyosin complex [1]. Adherent cells modify their behavior according to the substrate stiffness at a time scale of the order of 0.1 s [2]. This fast response is related to the cytoskeleton intrinsic rheological properties [3], whereas the mechanical response mediated by mechanosensing/mechanochemical signaling is unlikely, as its time scale is in the range from one to several minutes in repetitive stretching cycles [4,5]. In these experiments, straight stress fibers (SF) structures respond to external stimuli that ultimately allow adherent cells to change their orientation according to the imposed stress field.

The SFs are composed mainly of actin fibers bundle together with cross-linking proteins such as α-actinin, fascin, espin and filamin [6]. Furthermore, SF composition involves myosin-II molecular motors that allow the fibers to contract. However, in terms of structure, SF covers a wide range of possibilities, from poorly organized actomyosin bundles in fibroblast tissue to a highly organized sarcomere pattern in smooth muscle cells [6].

In terms of the SF mechanical response, laser ablation experiments on a single SF in living cells show that the viscoelastic-like retraction of the fiber depends on the contribution of myosin motors [7]. The severing of the fibers leads to the modification of the spatial organization of actin–zyxin complexes along adherent actin bundles [8], with a nonuniform spatial distribution along the SFs [9], in which the contraction is localized far from focal adhesions (FAs). Notably, the elongation of the SFs mainly occurs close to FAs [9] without any external stress. A mechanical description has been developed to capture the dynamics of the SF retraction after ablation. These models combine passive mechanical elements [7], such as springs and dashpot, with active elements that represent the molecular motor activity [8]. Attempts at an SF mechanical representation subdivide the bundle into smaller sections in order to account for spatial strain gradients [8,10,11], where each subsection does not differ in terms of the mechanical properties, from their neighbors. The active elements are based on the Huxley or hyperbolic model [12,13,14] for molecular motors, which is a particular case of the skeletal muscle that takes into account the actin–myosin interaction in a well defined and spatially organized structure [8,10,11].

Cellular response to mechanical stimuli has been studied extensively under a variety of approaches in order to observe the mechanical and biological effects of the applied forces [15]. Cellular rearrangements during stretch-induced cell polarization have been shown to induce a sliding of the focal adhesions, repolarization and reorganization in total independence of the microtubular network dynamics [16], the latter being essential for the FA stability [17].

However, the response of actin fibers due to mechanical solicitation has not yet been studied for a living cell in a noninvasive way. In this work, we study the dynamical response of actin fibers under stretch in F-actin-labeled retinal pigment epithelial cells, RPE-1 [18,19,20], plated on deformable, adhesive micropatterns (Figure 1A,B, Appendix A). The RPE-1 cell type is a well-suited cell line to study the organization of the actin cytoskeleton [18], its response under mechanical solicitation [20] and the biochemical response to geometrical constraints [19]. The PDMS sheet, used to stretch the cells, displays a negligible residual strain of the substrate in creep tests [21], ensuring that the observations correspond to the SF mechanical response. Moreover, the chosen micropattern allows us to obtain nonadherent peripheral SFs, located at the free edge of the cells [18,22] anchored to both extremes of the micropattern without sign of FAs detachment (Figure 1C). The relationship between the cytoskeleton activity and the radius of curvature at the free edge of a cell, described by Bischofs [23,24] allows us to compute the SF mechanical parameters, such as the SF line tension, the dissipation constant, the cortical tension and the motor protein contribution by comparing the experimental results and the proposed mechanical model of SF (Figure 1D). Our results attribute the crucial role in the cytoskeleton’s self-healing mechanism to the actomyosin complex. This is consistent with the idea of a large-scale mechanosensing mechanism for cellular adaptation to the stretch [25,26].

Then, in order to get more insight into the key mechanical features of SF under stretch, we study the response of the peripheral SF under compression and extension conditions by analyzing the radius of curvature at the free edge of the patterned cells, and also the intensity change of the fluorescent F-actin bundle over time at a global and local scale, in order to assess experimentally the features of the SFs and actin dynamics under an external mechanical solicitation.

### Stress Fibers’ Mechanical Model

In order to account for the SF mechanical response, we used an ad hoc Gaussian function based on the nonlinear response of motor protein experiments [27,28,29]. This Gaussian response presents two main features: the maximum tension value at a null rate of elongation and the typical velocity scale, which defines the Gaussian width of the molecular motor response. Furthermore, our model for protein motors is an average representation for the response of the ensemble of motors in the system, distributed in a spatially disordered cytoskeleton network. In the case of the Huxley or hyperbolic model [12,13,14] for molecular motors, there is a particular case for skeletal muscle, that takes into account only the actin–myosin interaction, in a well-defined and spatially organized structure. Nevertheless, both models of motor protein response, Gaussian and Huxley, share the same components. Therefore, both descriptions are equivalents. The main advantage of our approach is due to the Gaussian shape of the molecular motor representation, instead of a triangular function as in the case of Huxley. Last, we introduced computational challenges in order to solve the differential equation with discontinuous functions.

As shown in Figure 1D, the mechanical representation of the SF was achieved by combining passive and active elements. The temporal evolution of the SF under constant stretch is described by the equations: (1)$$\begin{array}{c} K\delta_{1}+\lambda_{0}=\lambda \end{array}$$(2)$$\begin{array}{c} k\delta_{2}+\gamma {\dot{\delta }}_{2}+\lambda_{0}e^{-{\dot{\delta }}_{2}^{2}/V^{2}}=\lambda \end{array}$$(3)$$\begin{array}{c} \delta_{1}+\delta_{2}=\delta l \end{array}$$

The mechanical parameters K and *k* stand for the main and the secondary elastic constants of the cytoskeletal elasticity. The viscous damping γ arises from the interaction of protein motors and biopolymers, due to the attachment/detachment dynamics driven by ATP hydrolysis cycle [29,30,31] and the initial line tension is denoted by $\lambda_{0}$. The motor protein average mechanical response is modeled by the element *M*, which consists of a Gaussian function with the maximum force $\lambda_{0}$ and the typical velocity scale *V* [30,31]. The values $\delta_{1}$ and $\delta_{2}$ are the elongations of the mechanical elements K and k−γ−M, respectively, and δl is the total elongation of the bundle. This mechanical representation is able to describe the extension and contraction of the SFs.

Considering that the main and secondary spring constant can change their values after the applied stretch, we can solve the set of Equations (Equation 1)–(3), at short and long time scales, in order to relate the radius of curvature of the SF as we describe below.

## 2. Results

As shown in Figure 1, at the cell edge, the F-actin SF forms an arc-segment of radius *R*, anchored to the SF adhesion sites. The SF under mechanical tension λ is curved by the cell cortical tension σ. Then, the SF radius of curvature is the mechanical balance between the line tension and the cortical tension, as R=λ/σ [23,24]. With this relation and the proposed model, we simplified the set of equations in order to understand the compression and extension experiments for early and long stages at a constant stretch as presented in Figure 2 (Materials and Methods section).

### 2.1. Static Analysis of Stretched Micro Patterned Cells

We measured the initial SF length and radii $L_{0}$ and $R_{0}$, respectively, for the chosen micropattern ($L_{0}=36.47\pm 4.21\;\mu$m and $R_{0}=65.60\pm 5.44\;\mu$m, Figure 1A, N = 30). The radius of curvature, immediately after the stretch is denoted by $R_{0}^{+}$, and the radius after 15 min of constant stretch by $R_{\infty }$ (Figure 2A,B). The SF radius temporal evolution is shown in Figure 2C, exhibiting a global relaxation time *T* over 300 s (〈T〉=327.3 s, SD=57.9 s, N = 64), which defines the time scale of the bundle response, with a total data acquisition time of 900 s for each experiment.

Due to the external mechanical stretch, it has been shown that zyxin is accumulated at the SF after stretch [32] accompanied by an increment of actin as well. These observations lead to the assumption of a rearrangement and reinforcement of the SF [33] that could involve changes in the elasticity of the SF. Therefore, we considered that the main and secondary spring constant value depended on the amount of the applied stretch with a time scale that governed this dynamics. Thus, in a more general description, the elastic constant of the SF is: (4)$$\begin{array}{c} K=K_{0}+F\left(\epsilon ,t\right) \end{array}$$(5)$$\begin{array}{c} k=k(\epsilon ,t) \end{array}$$
where $K_{0}$ is the main elastic constant of the SF at null strain, F(ϵ,t) is a general function that describes the response of the mechanics of the SF under mechanical solicitation and k(ϵ,t) is the secondary spring whose value also depends on the applied strain.

#### 2.1.1. Short Time Scale Analysis

After deformation, in both extension (Appendix A) and compression (Appendix A) cases, the SFs behave elastically as a function of the applied strain ϵ, as shown in Figure 2D. In extension (black circles), the radius of curvature difference, $R_{0}^{+}-R_{0}$, displays a slope equal to 189.5±16.6μm, while in compression (black squares), the radius of curvature difference with respect to the applied strain is 177.4±23.7μm, with a relative difference between compression and extension of approximately 6%. Note that in the compression case, the SFs are initially prestretched for 60 min (Figure 2A,C, bottom panel). Then, the imposed prestretched radius of curvature is compressed (Figure 2B, bottom panel). Therefore, the applied strain, measured from $R_{0}^{p}$, is negative. This assumption has been demonstrated to hold true in cyclic stress experiments in fibroblast, showing that cells return back to a homeostatic state within 90 min of constant stretch [4,5]. Furthermore, we did not observe any disruption or buckling of the SF during the compression procedure. In fact, to observe buckling in SF requires instantaneous compression at compression rates of 26% in less than five seconds [34].

This linear response of the SF to the applied strain can be explained by the intrinsic elasticity of the cell cytoskeleton, and more importantly, by the time required for the mechanotransduction pathways to react. Then, the main spring constant at a short time scale, Equation (Equation 4), becomes $K{\left(\epsilon \right)}_{t=0}=K_{0}$.

From the proposed model, at compression and extension conditions, the main spring constant is $K_{0}$. Then, for early stages of elongation (t ≪T), it holds that $\delta_{1}\approx \delta l=L_{0}\epsilon$, where $\epsilon =\Delta L/L_{0}$. Then, the Equation (Equation 1) becomes $\lambda =K_{0}L_{0}\epsilon +\lambda_{0}$, with $L_{0}$ as the initial actin bundle length. Combining the modified expression of λ and R=λ/σ, leads to the description of the radius of curvature immediately after the stretch as:(6)$$R_{0}^{+}=\left(\frac{K_{0}L_{0}}{\sigma }\right)\epsilon +R_{0},$$
whose slope is $K_{0}L_{0}/\sigma =189.5\pm 16.6\;\mu$m and $R_{0}=\lambda_{0}/\sigma =66.2\pm 4.2\;\mu$m as the nonstretched radius of curvature, with a linear dependency on the strain as shown experimentally for both compression and extension cases (Figure 2D).

#### 2.1.2. Long Time Scale Analysis

After five minutes of constant stretch, the evolution of the SF radius of curvature reached a plateau and we were able to observe the differences between the compression and extension experiments. Indeed, the initial consideration related to SF modeled as the combination of viscoelastic and active objects [8,9,10] and the experimental data indicated that this was not the case.

In the compression experiments, the SFs relaxed as a Maxwell material, i.e., without the participation of a secondary spring constant. Thus, in compression, the secondary spring, (5), could be considered as $k{\left(\epsilon ,t\right)}_{\epsilon <0}=0$. This assumption is based on the temporal evolution of the SF shown in Figure 2C. The SF after compression reaches a mean value of $R_{\infty }=65.1\pm 5.2\;\mu$m, independently of the applied compression strain, and the average radius of curvature difference at short and long time scales $\left(R_{\infty }/R_{0}-1\right)$ is less than 2%. In this case, the relaxation implies that $\dot{\delta }l={\dot{\delta }}_{1}={\dot{\delta }}_{2}=0$, which corresponds to the theoretical solution given by $R_{\infty }=R_{0}=\lambda_{0}/\sigma$ for ϵ<0 (Figure 2D, red squares). The characteristic time scale of the process is of the order of 300 s, Figure 2B bottom panel, suggesting that the motors are pushed but not compressed, gliding back the SF to the initial radius of curvature involving an effective viscosity [7,29], as the presented result is far from the required compression rate to observe buckling [34].

For extension conditions, the SF behaved like a more complex material (Figure 2D, red circles), where the radius of curvature $R_{\infty }$ decreased as in the case of a viscoelastic material, which has been suggested previously [8,9,10]. We thus assumed that the main spring value was modified by the applied strain and the secondary spring could only be tested at positive strain values. Then, for a longer time scale at positive strain Equations (Equation 4) and (5) can be written as $K{\left(\epsilon \right)}_{t=\infty }=K_{\infty }$ and $k{\left(\epsilon >0\right)}_{t=\infty }=k$.

The data obtained for a larger time scale (t>T) indicated that ${\dot{\delta }}_{2}\approx 0$. The values of the main and secondary spring were $K_{\infty }$ and *k*, respectively. These considerations implied that Equation () became $\lambda =k\delta_{2}+\lambda_{0}$. By combining Equations (Equation 1)–(3), we obtained the line tension equal to $\lambda =kK_{\infty }/\left(k+K_{\infty }\right)L_{0}\epsilon +\lambda_{0}$. Dividing this expression by σ, the SF radius of curvature, at t→∞, led to $R_{\infty }=kK_{\infty }/\left(k+K_{\infty }\right)\left(L_{0}/\sigma \right)\epsilon +\lambda_{0}/\sigma$ for ϵ>0, with an experimental slope equal to 104.9±25.5μm (Figure 2D, red circles, the red segmented line represents the solution of Equation (Equation 9)).

The results at long time scale give us the opportunity to describe more precisely the model in terms of strain. Indeed, the experiments suggested a significant difference in the cytoskeleton reorganization process between these two cases, that can be expressed as:(7)$$K\left(\epsilon ,t\right)=\{\begin{array}{ccc} K_{0}\; & ,\;\epsilon <0 & \\ K_{0}+F\left(\epsilon ,t\right) & ,\;\epsilon >0 \end{array}\;k\left(\epsilon ,t\right)=\{\begin{array}{ccc} 0 & ,\;\epsilon <0 & \\ k & ,\;\epsilon >0 \end{array}$$

### 2.2. Fresh Actin Recruitment during Compression and Extension of Actin Bundles

The mechanical properties of SFs depend on its molecular composition and structure. Considering that the F-actin is the main actor responsible of the SF elastic features [35], we analyzed the total fluorescent label actin intensity assuming that the total fluorescence intensity was proportional to the amount of F-actin in the SF [11]. The F-actin fluorescence intensity analysis yielded different outcomes for the compression and extension experiments.

In the extension experiments, the total SF F-actin intensity *I* remained constant during the extension procedure (Figure 3A gray region, t<0) and increased immediately after the extension procedure stopped (Figure 3A top panel, t>0). After 300 s of constant stretch, the bundle actin intensity reached a steady-state value, denoted with the increment ΔI, indicating that new actin units were being recruited in the SF, with a slope denoted by α (Figure 3B, ϵ>0). Therefore, the plateau value as a function of the applied strain was ${\left(\Delta I/I_{0}\right)}_{t=\infty }=\alpha \epsilon$. Furthermore, at the shorter time scale, the initial rate of actin recruitment, shown in Figure 3C, also depended on the applied stretch as $d/dt{\left(\Delta I/I_{0}\right)}_{t=0}=\left(\alpha /t^{*}\right)\epsilon$. From the experimental data, in extension conditions (Figure 3B,C, ϵ>0), we obtained α=0.64±0.15 and the experimental rate of actin recruitment was $\alpha /t^{*}=\left(1.5\pm 0.2\right)\times 10^{-3}$ s${}^{-1}$. Thus, the typical time scale for this process was $t^{*}=427\pm 43$ s, allowing us to model the intensity variation as $\Delta I/I_{0}=\alpha \epsilon \left(1-e^{-t/t^{*}}\right)$ s. This last expression is only valid for extension conditions. This result suggests that the rate of actin recruitment is dynamically modulated in the stretched SFs.

In the compression experiments, the cells were initially prestretched and maintained stretched for 60 min. Subsequently, the amount of F-actin fluorescence intensity increased to a new value, $I_{0}+\Delta I_{p}$ (Figure 3A). The new intensity value remained constant during the compression procedure and, more importantly, did not show any notable evolution afterwards (Figure 3B, ϵ<0). For the explored range of negative strain (approximately 30%), we did not observe a decrease of the F-actin signal. This may indicate that the rates of polymerization/depolymerization were equilibrated. Only in extreme cases of 80% of compression did the retraction of the SFs lead to a high F-actin overlap, accompanied by an actin disassembly increase of 50% after a critical density value was reached [11].

Computing the intensity difference, we showed that we did not come back to the same density value, defining the linear density of F-actin intensity as ρ=(total intensity/SF length). For a purely elastic material, the linear density variation should evolve in a linear fashion with respect to the applied strain as $\rho_{\infty }/\rho_{0}-1=-\epsilon$ (Figure 3D, red segmented line), where $\rho_{\infty }$ is defined as the average intensity value over the SF length at t→∞ and $\rho_{0}$ defined at t=0.

The relative linear density increased with the amount of compression ${\left(\rho_{\infty }/\rho_{0}-1\right)}_{\epsilon <0}$=−(1.14±0.67)ϵ (Figure 3D, solid red line ϵ<0). Indeed, the prestretch procedure set the actin level in the SF to a new and higher value, which remained constant through the compression experiment (Figure 3A) while the radius of curvature and the length of the SF reverted to their unstretched value (Figure 2D). As a result, there was an increase in the SF actin density, which is explained by the compression and gliding of the actin SF at constant protein content. This observation is in agreement with the increase of actin density reported on the artificially severed SFs [8], with linear density doubling after SF retraction followed by a depolymerization of the bundle [11].

For the extension experiments, the SFs reacted to the applied strain by recruiting more actin into the bundle (Figure 3B). The slope of the density variation differed from the compression case, with an experimental linear density variation ${\left(\rho_{\infty }/\rho_{0}-1\right)}_{\epsilon >0}=-\left(0.64\pm 0.04\right)\epsilon$ (Figure 3D, red solid line ϵ>0). In the hypothetical case of a total recovery of the SF, the density variation slope could reach a null value (Figure 3D, red dotted line with α≫1), meaning that the SF was maintained at a constant density equal to $\rho_{0}$. In the purely elastic case, no density variation should be noticed in extension nor compression conditions (Figure 3D, red segmented line with α=0). The phenomenon observed on the radius of curvature at the longer time scale (Figure 2D) is explained by the recruitment of the new actin material into the SF. This new actin material changes the density of the SF, and more importantly the SF mechanical features [36,37,38,39]. This is in agreement with our measurement of the radius of curvature of SF under stretch. It suggests that the mechanism of reorganization of actin bundles is different for compression conditions (process at a constant F-actin protein content) and extension conditions (process that incorporates new actin proteins).

Assuming that the fluorescence intensity variations of the labeled F-actin (Figure 3A) act similarly in the variations of the SF elasticity [4], the general function F(ϵ,t) is proportional to the applied strain and reaches a steady-state value at the long time scale. Since K(ϵ,t) is reduced to $K_{\infty }$ at the long time scale, the unknown function becomes: $F{\left(\epsilon \right)}_{t\to \infty }=-\alpha K_{0}$. Then, the temporal dependency of K(ϵ,t), due to the addition of fresh actin units, can be written as: $K\left(\epsilon ,t\right)=K_{0}-\alpha K_{0}\epsilon \left(1-e^{-t/t^{*}}\right)$, with $t^{*}$ as the time scale of the main spring variation, whose steady-state value is given by $K_{\infty }=\left(1-\alpha \epsilon \right)K_{0}$. Then, α stands for the index of addition of fresh actin units to the bundle in response to the applied stretch and $\alpha /t^{*}$ stands for the rate of protein recruitment (Figure 3B,C). Therefore, the addition of the fresh actin into the SF, preventing the spontaneous severing of the SF, brings as a consequence the softening of the actin bundle [36,37,38], with the reduction of the main elastic component to $\Delta K/K_{0}=-\alpha \epsilon$, leading to a reduction of 19% for a strain value of ϵ=0.3.
(8)$$K\left(\epsilon ,t\right)=\{\begin{array}{ccc} \;K_{0} & ,\;\epsilon <0 & \\ K_{0}-\alpha K_{0}\left(1-e^{-t/t^{*}}\right) & ,\;\epsilon >0 \end{array}\;k\left(\epsilon ,t\right)=\{\begin{array}{ccc} 0 & ,\;\epsilon <0 & \\ k & ,\;\epsilon >0 \end{array}$$

The decrease of the main elastic constant of the SF brings as a consequence that the response of the SF radius of curvature, at a long time scale, is not the same as that of a viscoelastic material. Indeed, in the static analysis section, we found that the radius of curvature variation was $\Delta R_{\infty }=kK_{\infty }/\left(k+K_{\infty }\right)\left(L_{0}/\sigma \right)\epsilon$. Thus, using the expression for the main elastic constant steady state value $K_{\infty }=\left(1-\alpha \epsilon \right)K_{0}$ leads in the first-order approximation to a quadratic dependency of the radius of curvature on the applied extension strain at t→∞ as:(9)$$\Delta R_{\infty }=\frac{kK_{0}}{k+K_{0}}\;\frac{L_{0}}{\sigma }\;\left(1-\alpha \frac{k}{k+K_{0}}\epsilon \right)\epsilon ,$$
as shown in Figure 2D (red circles, the segmented red line represents the theoretical solution without adjustable parameters).

### 2.3. Global to Local Dynamics of Stretched Actin Bundles

To understand the global dynamics shown in Figure 3, we explored the F-actin activity at a local scale when the SFs were subjected to stretch. As shown in Figure 4, the stretched actin SF kymograph is complex (Figure 4B). First, high actin concentrations are present at both ends of the SF, which correspond to the focal adhesion sites (FAs, Figure 4B,C red circle, and Figure 4D). Second, almost 100% of the SF displays less actin abundance at the middle of the SF (mSF, Figure 4B,C red square, and Figure 4D). As shown in Figure 4C, these features are maintained throughout the experiment at a constant stretch.

The temporal evolution shows an overshoot at FA regions (Figure 4C red circle, and Figure 4E), whose time scale is in the order of $\tau_{\mathrm{FA}}=178.5\pm 28.3$ s (Figure 4F). Then, the actin signal decreases, reaching a plateau after five minutes. At the middle of the SF, the temporal evolution displays that in the mSF surroundings the actin is accumulated, reinforcing the “weakest” location of the SF (Figure 4C, red square, and Figure 4G). This process does not show any evident intensity overshoot, but a steady increment instead, reaching a plateau with a time scale in the order of $\tau_{\mathrm{mSF}}=387.9\pm 23.2$ s (Figure 4H). Because of the two different time scales, these results describe two different pathways of actin polymerization.

### 2.4. Local Temporal Evolution Analysis of the SF under Stretch

Notice that the local curvature and intensity variation show identical dynamics (Figure 4F), while the global radius of curvature (Figure 2B) and the total intensity along the SF (Figure 3A) displays an exponential-like evolution ($R^{-1}\propto \Delta I/I_{0}$). Furthermore, at a given segment of the SF, the line tension is the same at both ends of the segment that can be represented by the mechanical device shown in Figure 1D. Thus, at a constant stretch $\dot{\delta }l\approx 0$, ${\dot{\delta }}_{1}=-{\dot{\delta }}_{2}$. This condition reduces the complexity of the set of equations of our model. To simplify the problem, we considered that at a short time scale, the radius of curvature behaved elastically (Figure 2D). The SF radius of curvature displayed small variations under constant stretch, less than 15% for a strain value of 0.4. The main elastic constant varied up to 20% for a strain value of ϵ=0.3. The F-Actin density was not far from the elastic case, Figure 3D. Thus, we assumed in the subsequent analysis that the mechanical properties did not vary and could be considered constant through the time course of the experiment.

Then, to study the local dynamics of the SFs under stretch, we used the fact that the curvature, the line tension and the cortical tension satisfy the mechanical balance [23] at a global and local scale. Then, Bischofs’s relation holds true also at a local scale $\sigma_{l}=\lambda_{l}/r_{l}$, where $\sigma_{l}$ , $\lambda_{l}$ and $r_{l}$ are the cortical tension, line tension and radius of curvature at a local scale. Expressing the local radius of curvature as $r_{l}=R_{0}+\delta r$, we can solve the set of Equations (Equation 1)–(3) applied to the F-actin intensity kymograph data with local mechanical parameters (Materials and Methods section), which leads us to:(10)$$\ddot{\chi }\left(\Gamma +\dot{\chi }e^{-{\dot{\chi }}^{2}}\right)=-\nu f+\nu^{2}\Gamma \chi +\nu \left(1+\dot{\chi }e^{-{\dot{\chi }}^{2}}\right)$$

Differential Equation (Equation 10) allows us to extract mechanical information directly from the normalized intensity kymographs (Figure 4B) at any location within the SF due to the relation between curvature and fluorescent intensity [11] (Figure 4E,G). The fit parameters, for both compression and extension experiments, were obtained by comparing the numerical solution of Equation (Equation 10) and the experimental intensity profiles [8,11] by a least-squares minimization as described in the Materials and Methods section.

The values obtained from the local kymograph analysis depended on the length of the bundle segment used for averaging the fluorescent signal, the latter being performed to reduce the signal fluctuations. From the analysis of the parameters obtained, fitting the intensity profile data to our model, we found a characteristic segment length between 1 and 10μm. The upper limit could be obtained by analyzing the parameter $f=K_{0}\epsilon /2\sigma_{0}∼1$. Indeed, considering the slope of the initial elastic behavior for the chosen pattern, $K_{0}L_{0}/\sigma_{0}\approx 200\;\mu$m, the elastic modulus [40] of the SF ($E=11.3\times 10^{3}$ Pa), and the main elastic constant ($K_{0}\approx EA/L$), led to $f∼\left(EA/\sigma_{0}\right)\left(\epsilon /2L\right)=1$, then L, the unknown length scale, was approximately of the order of 10μm. For a larger value of the parameter *f*, the length scale L decreased up to approximately 1μm. On the other hand, considering the structure and protein distribution of the SF reported in the literature, the minimum length scale could be associated to the separation of zyxin proteins, which is on the order of 1μm [8,9,11].

Averaging the relative intensity profiles, using 5μm length segments (Figure 4B segmented rectangular regions), we found that the initial cortical tension $\sigma_{0}=35\left(SD\;15\right)$ pN/μm and the initial line tension $\lambda_{0}=1.9\left(SD\;1.4\right)$ nN were in the range of values obtained from the analysis of the curvature radius at short and large time scales (static analysis of stretched micropatterned cells section).

The viscoelastic time scale $\nu^{-1}$ was observed in the 200–500 s range due to the actin and motor proteins interaction, as in the case of the contraction time scale observed in severed SF [7]. Otherwise this time scale, $\nu^{-1}$, should be on the order of one second [7]. On the other hand, the active time scale τ, due to motor protein activity was in the range of 100–300 s. The competition between motors Γ (sliding motors and motors exerting forces) was in the range of 0.3–1.4. The elasticity and the cortical stress ratio *f* ranged between 2 and 10. As a comparison, theoretical values derived from the model showed that the viscoelastic time scale $\nu^{-1}$ was on the order of 300 s and the time scale associated to motor activity τ was on the order of 200 s. The elastic to the cortical tension ratio *f* was on the order of 1, and the competition between motors Γ∼1.

## 3. Discussion

In this paper, we used the formation of peripheral stress fibers [18], located at the free edge of RPE-1 cells plated onto H-shaped micro-pattern, to obtain curved SF as a result of the cell cortical tension and the SF linear tension, R=λ/σ. The limitation of this study was to rely on anchor actin bundles onto the extracellular matrix. Another approach should be used to study other type of SFs such as actin bundles in wound healing [41] or actin bundles developed between cells in epithelium [42]. Our approach allowed us to directly and nonintrusively obtain information on the mechanics of the SFs by relating the radius of curvature over time to its mechanical response (Figure 1 and Figure 2).

From our theoretical analysis and the experimental data, we established that the experimental slope in Figure 2D was related to $K_{0}L_{0}/\sigma$, Equation (Equation 6). Then, considering that the extensional spring constant $K_{0}=EA/L_{0}$ (where E and A are the Young’s modulus and the cross-sectional area of the material, average elastic modulus of SF in living cells $E=11.3\times 10^{3}$ Pa = 11.3 nN/μm${}^{2}$ [40], and average cross-sectional area of the peripheral SF of RPE-1 cells SF 〈A〉=0.6μm${}^{2}$, SD = 0.2μm${}^{2}$, N = 64), led to the average cortical tension of the cell as $K_{0}L_{0}/\sigma =EA/\sigma =189.5\;\mu$m, obtaining an average cortical tension of 〈σ〉=35.7pN/μm (SD = 11.9 pN/μm, N = 64), which was comparable to the cortical tension found on the growth of membrane blebs of filamin-deficient melanoma cells σ≈55 pN/μm [43]. Furthermore, with the results obtained in the short and long time scale analysis, we were able to compute the ratio between both elastic constants, at small strain, which was found to be on the order of $K_{0}/k\approx 0.8$, leading to the average initial main and secondary elastic constants to be $K_{0}\approx 0.19$ nN/μm and k≈0.23 nN/μm, respectively. In contrast, in the case of PC12 neurites, the elastic constant ratio is of the order $K_{0}/k\approx 20$ [30]. This difference in the ratio of both elastic constant indicates that the mutually aligned actin filaments interacting with molecular motors form a dense network in RPE-1 cells in contrast to the neurite case [30,31,44,45]

Furthermore, taking into account Bischofs’s model, we were able to establish that the elasticity to cortex tension ratio *f* was related to the fiber rigidity to surface tension parameter [23] $l_{f}=EA/\sigma$. Thus, according to our data, we estimated that RPE-1 cells exhibited a $l_{f}$ value in the range of 100 to 200μm and an estimation of the dimensionless rest-length parameter in the order of 0.6 to 0.75, which is comparable to Bischofs’s findings considering a cable network elasticity control model [23].

The average initial line tension was then obtained from the cortical tension and the unperturbed SF radius of curvature, $R_{0}=\lambda_{0}/\sigma$, leading to an average line tension $\langle \lambda_{0}\rangle =2.3$ nN (SD = 0.8 nN, N = 64), which is comparable to the local measurements presented in Table 1. This line tension is in the range of the line tension deduced from references [9,46]. Thus, from a simple stretching experiment and by analyzing the SF radius of curvature, at a short time scale, it is possible to extract the cortical tension and the line tension along the SF of micro-patterned cell.

Our results on the SF stretching at a long time scale showed a clear difference in contrast to the case of a Hookean material (Figure 2). In compression, a slow recovery of the SF radius of curvature was observed (Figure 2D, ϵ<0). The SFs glided back to the initial radius of curvature without any elastic contribution of the cross-linkers, as in the case of a Maxwell material (k(ϵ)=0), while the SFs actin density increased as in the case of artificially severed SFs [11] (Figure 3D, ϵ<0). In extension, the SFs displayed an increment in the actin signal as a function of the applied strain (Figure 3B,C). The addition of the new actin material gave, as a consequence, a lower actin density after stretch (compared to the initial density), but higher than in the case of a pure elastic material (Figure 3D, ϵ>0).

The F-actin fluorescence intensity analysis revealed a change in the SF content modifying the elasticity of the bundle. Our measurements of the radius of curvature suggested that the F-actin bundle reacted to the applied strain by modifying the elasticity of the SF. This modification was due to the disruption of the actin cross-linkers and to the incorporation of unstretched fresh material into the bundle [32]. Therefore, the more the SF was stretched, the more actin nucleation sites were available (Figure 3B). Then, the rate of actin recruitment was only related to the polymerization rate given by the parameter $\alpha /t^{*}$ (Figure 3C). The incorporation of the fresh material into the SF under stretch reduced the elasticity of the actin bundle according to the result expressed by the Equation (Equation 8), explaining the radius of curvature difference in extension, Equation (Equation 9), shown in Figure 2D (red circles and red segmented line).

We showed that the heterogeneity of the protein distribution along the SF first led to variations of the local mechanical properties (Figure 2D), which has been suggested in isolated peripheral SF [47]. Second, it triggered different responses under stretch incorporating new actin material at the mSF and FA locations (Figure 4D,F,H). Third, the local intensity of F-actin at FA sites increased in extension. This observation is in agreement with the mechanotransduction of the FAs in the generation of new actin nucleation sites at the FAs under strain [48]. The actin intensity response of these locations (Figure 4E) displayed an overshoot of the actin signal, meaning that the rate of polymerization overcame the rate of depolymerization due to the applied stretch [49], with a time scale of these rates on the order of $\tau_{\mathrm{FA}}=180$ s (Figure 4F). This time scale is similar to the time scale of the mechanotransduction pathway of the FA under stretch [50] and is in the same order of magnitude as the active time scale τ in our model.

At the middle section of the actin bundle (mSF), we observed an increment of the actin intensity with a typical time scale of $\tau_{\mathrm{mSF}}=390$ s as shown in Figure 4H and it was in the same range of values as $\nu^{-1}$ contained in our model. This phenomenon takes place far from the regulatory role of the FAs. The new actin proteins are recruited and incorporated to the SF as shown by Smith et al. [32], where zyxin and VASP proteins mediate the incorporation of actin in order to maintain the mechanical homeostasis of the SF. This last feature has also been observed in microtubules, where mechanical stress [50] is responsible for a self-healing process [51,52].

## 4. Materials and Methods

### 4.1. PDMS Micro Patterning

The micropatterning procedure was based on Azioune’s protocol [22] and adapted by Manuel Théry to PDMS substrates [18]. PDMS sheets of 3 cm × 2 cm × 150 μm (Gel-Pak Hayward, CA, USA) were photoactivated by deep UV light exposition (180 nm) for five minutes. Immediately after that exposition, EDC/Sulfo-NHS was incubated for 15 min and rinsed with DH2O, followed by an incubation of PLL-g-PEG at 0.5 mg/mL in Hepes buffer 10 mM at pH 8.6 for three hours or over night. At this stage, the entire PDMS surface was passivated and, which prevented cell adhesion. To achieve the patterning of the PDMS sample, the substrate was transferred to the photomask, which contained the selected pattern, to be subsequently exposed to a deep UV light for five minutes. At this step, the deep UV light destroyed the exposed PLL-g-PEG polymer and activated the affected area. Finally, a fibronectin solution at 25μg/mL in NaHCO${}_{3}$ 100 mM at pH 8.6 was incubated for one hour at room temperature.

### 4.2. Cell Culture and Experiments

A culture of RPE-1 cells was prepared under standard protocols using Gibco Leibovitz’s L-15 medium with a HEPES buffer, 10% of fetal bovine serum and antibiotics obtained from Invitrogen. In these experiments, the substrate was placed in the stretcher apparatus [20,22,53], where the main component was a PI-Line M-663 displacement motor (Physik Instrumente, Germany). Cells were plated in a 1 cm${}^{2}$ PDMS frame placed on the micropatterned PDMS sheet with a concentration of 1000 cells/cm${}^{2}$, using a Leibovitz’s L-15 medium with no phenol red to avoid fluorescent background. After 15 min, cells started to adhere to the patterned surface. Nonadhering cells were removed by a gentle aspiration. A cover glass was used to seal the PDMS frame to prevent evaporation and changes of the medium pH. The microscope was thermally stabilized at 37 ${}^{\circ }$C with a Plexiglas chamber that enclosed the setup. A temperature controller (Life Imaging Services, Switzerland) was used to perform long-term experiments. After seeding the RPE-1 cells into the chamber, a full spreading of the cells was reached within sixty minutes, and two hours later, the F-actin (labeled with Red Live Actin) network was organized into actin fibers in the cytoplasm and actin bundles at the free edge of cells onto H-like PLL-g-PEG/fibronectin PDMS patterns. At this stage, RPE-1 cells were being stretched up to 40% of their initial length at a rate of 100μm/s, to ensure that the initial cell response was mainly elastic. Plated cells were recorded before and after the stretching procedure.

### 4.3. Image Acquisition and Analysis

After stretching, cells were recorded for fifteen minutes, acquiring images every two seconds for the dynamic response study. The time delay before stretching and the first image was between ten and fifty seconds depending on the amount of stretch. The fluorescence images were recorded by using an EZ-CoolSnap Photometrics camera (Tucson, AZ, USA) and a Nikon Eclipse-Ti microscope. The image centering was achieved using ImageJ software and the TurboReg plugin. The photobleaching intensity correction, the kymographs, the relative intensity profiles, the local and global radii of curvature and the numerical fits were computed using Matlab software (MathWorks, Natick, MA, USA). The photobleaching was corrected by using the exponential decay method described in [54]. The kymographs were computed using the relative normalized intensity. This analysis shows the relative variation of intensity in a selected region. In Figure 2 and Figure 3 we used the standard deviation for the error bar. Unless otherwise mentioned, the number of stress fibers was equal to the number of cells.

From the linear relationship between the relative change of the fluorescent signal and the variations in the cortical tension [11], the fitting procedure of the intensity profiles corresponded to an optimization of the set of parameters using a least-squares minimization of the model and the experimental data, with the experimental value of the parameters k/K, σ, λ and *f* obtained from the strain-radius data analysis and using the first and second experimental derivatives of the normalized intensity profiles as inputs and initial conditions for the numerical integration of Equation (Equation 10).

### 4.4. Dynamics under Stretch

Considering that the relative variations of intensity are proportional to the curvature of the SF (Figure 4F) and the Bischofs relation, σ=λ/R, we are able to give a description of the temporal evolution of the SF in terms of the relative change of the cortical tension as follows. At a local scale, the radius of curvature, the line tension and the cortical tension also satisfy the balance $\sigma_{l}=\lambda_{/}r_{l}$ (the subscript *l* accounts for a local description). At a given location along the bundle, the mechanical equilibrium is described by Equations (Equation 1)–(3), where δl is now a local elongation in the SF, and $\epsilon_{l}\approx \delta r/R_{0}$ is the local strain. Therefore, the local strains of the mechanical elements are $\epsilon_{1}$ and $\epsilon_{2}$. Then, the set of equations becomes:(11)$$\begin{array}{c} K\epsilon_{1}+\sigma_{0}=\sigma_{l} \end{array}$$(12)$$\begin{array}{c} k\epsilon_{2}+\gamma {\dot{\epsilon }}_{2}+\sigma_{0}e^{-{\dot{\epsilon }}_{2}^{2}R_{0}^{2}/V^{2}}=\sigma_{l} \end{array}$$(13)$$\begin{array}{c} \epsilon_{1}+\epsilon_{2}=\epsilon_{l} \end{array}$$

Due to the nonlinear nature of the Gaussian model for the molecular motor contribution used here (Equation (2), and considering that the relative change of the main spring constant under constant stretch (at t→∞) is: $\Delta K/K_{0}=-\alpha \epsilon$, which can be reduced up to 25% for a high strain value ϵ∼0.4, we consider that the elastic constant is a constant parameter. Then, at constant stretching, the local strain $\epsilon_{l}$ remains constant, thus ${\dot{\epsilon }}_{l}=0$ and ${\dot{\epsilon }}_{1}=-{\dot{\epsilon }}_{2}$. Using the new expression for Equations (Equation 1) and (2), leads to:(14)$$k\left(\epsilon_{l}-\epsilon_{1}\right)-\gamma {\dot{\epsilon }}_{1}+\sigma_{0}e^{-{\dot{\epsilon }}_{1}^{2}R_{0}^{2}/V^{2}}=\sigma_{l}$$

Computing the time derivative of Equation (Equation 11) ($K_{0}{\dot{\epsilon }}_{1}={\dot{\sigma }}_{l}$), and denoting $\xi =\left(\sigma_{l}-\sigma_{0}\right)/\sigma_{0}=\Delta \sigma_{l}/\sigma_{0}$, we obtain a differential equation that describes the time evolution of the relative variation of the cortical tension, ξ, as:(15)$$\frac{\gamma }{K_{0}}\dot{\xi }+\left(1+\frac{k}{K_{0}}\right)=\frac{k\epsilon_{1}}{\sigma_{0}}\left(1-e^{-\frac{\sigma_{0}^{2}\;R_{0}^{2}}{K_{0}^{2}\;V^{2}}{\dot{\xi }}^{2}}\right),$$
with $\tau =\sigma_{0}R_{0}/K_{0}V$ as a typical active time scale. Considering $k/K_{0}∼1$ (results in the static analysis of stretched micropatterned cells) and using the change of variable ξ=χ/τ, Equation (Equation 15) becomes:(16)$$\dot{\chi }=\left(\frac{K_{0}R_{0}}{\gamma V}\right)-\frac{2K_{0}}{\gamma }\chi -\left(\frac{\sigma_{0}R_{0}}{\gamma V}\right)\left(1-e^{-{\dot{\chi }}^{2}}\right)$$

Finally, taking a derivative with respect to time of the Equation (Equation 16), we obtain a second order nonlinear differential equation for the scaled relative change in the effective cortical tension, as:$$\ddot{\chi }\left(\Gamma +\dot{\chi }e^{-{\dot{\chi }}^{2}}\right)=-\nu f+\nu^{2}\Gamma \chi +\nu \left(1+\dot{\chi }e^{-{\dot{\chi }}^{2}}\right)$$
with the parameter $\nu^{-1}=\gamma /2K_{0}$ as the classical relaxation time scale of a damping system. The time scale, $\tau =\sigma_{0}R_{0}/K_{0}V$, is due to the motor activity that pulls on actin filaments. The initial elastic behavior is captured by the parameter $f=K_{0}\epsilon_{l}/2\sigma_{0}$ , which represents the ratio between the elastic and the cortical tension. Finally, a dimensionless parameter, $\Gamma =\gamma V/2\sigma_{0}R_{0}$, takes into account the competition between protein motors exerting forces (active contraction) and those motors that slide between fibers due to shear forces (viscosity).

## 5. Conclusions

Since both actin processes, at FAs and mSF, are at different time scales, the presence of an overshoot could be explained by the simultaneous action of the mechanotransduction at FA locations and the healing process at mSF. As a sketch, pulling on the SF triggers the response of the FAs, and at the same time, the mSF is repaired, influencing the sensing of the strain by the FAs. Thus, the mechanotransduction and self-healing are responsible for the SFs mechanical integrity in extension. This response is more complex than the response observed in grown actin bundles submitted to stretch [52]. Indeed, the lack of cross-link proteins, the ability to incorporate fresh actin material into the bundle or feedback of any kind result in a classical Maxwell behavior.

The response of SFs under mechanical solicitation transits from actin fiber gliding along the bundle in compression to actin polymerization and actin rearrangement in extension [47]. As a consequence, the SFs now display strain-softening features, which help to prevent the spontaneous severing of stress fibers under stretch.

## Figures and Tables

**Figure 1 ijms-23-05095-f001:**
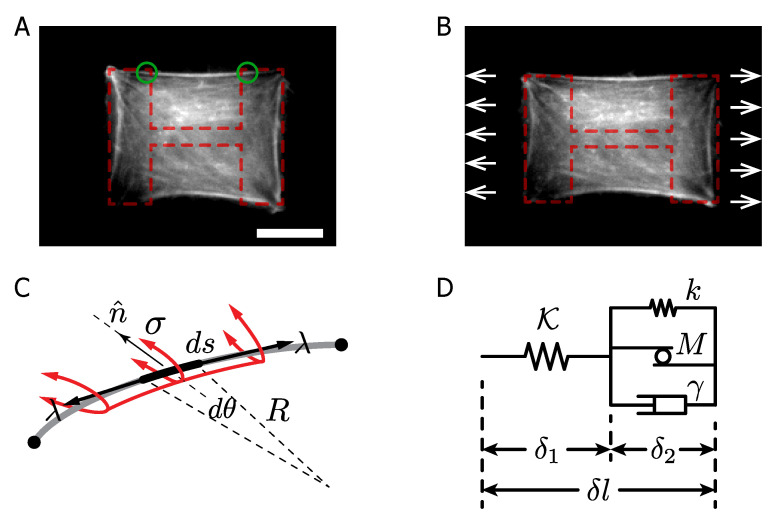
F-actin-labeled RPE-1 cells on PLL-g-PEG/fibronectin patterns. (**A**,**B**) Cells with main F-actin bundles at the free edge before stretch and the same cell after 15 min at a constant stretch (H-like pattern is displayed on segmented red lines, while the green circles target the FAs location). The experimental apparatus stretches the PDMS substrate, exerting tension on the cell and the stress fibers as well. Scale bar 20μm. (**C**) Sketch of an F-actin bundle between two anchored points (FAs), with radius of curvature *R*, cortical tension σ and the line tension λ along the bundle. (**D**) Mechanical representation of the SF.

**Figure 2 ijms-23-05095-f002:**
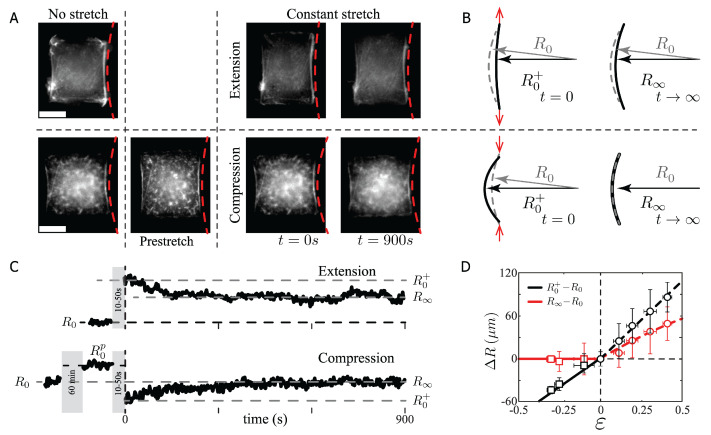
Radius of curvature of SFs vs. the applied strain. (**A**) Two examples of RPE-1 cells plated on H-like patterns before and during the applied stretch. Scale bar 20μm. (**B**) Sketch of the SF and the definition of the radius of curvature. (**C**) Response of the radius of curvature as a function of time for extension and compression conditions. Note that the initial radius of curvature for compression experiments corresponds to a prestretched state with initial radius $R_{0}^{p}$, whereas $R_{0}^{+}$ and $R_{\infty }$ correspond to the radii just after stretch and after fifteen minutes of constant stretch, respectively. The gray region represents the delay on the observation due to the stretching process at a velocity of 100μm/s. (**D**) Radius of curvature difference at short and long time scale (circles and squares symbols) for extension and compression experiments. N = 12 SF for each data point. Notice that the error bars amplitude, in all plots, correspond to the standard deviation (SD); therefore, the uncertainty of the data correspond to the SD value divided by $\sqrt{N}$.

**Figure 3 ijms-23-05095-f003:**
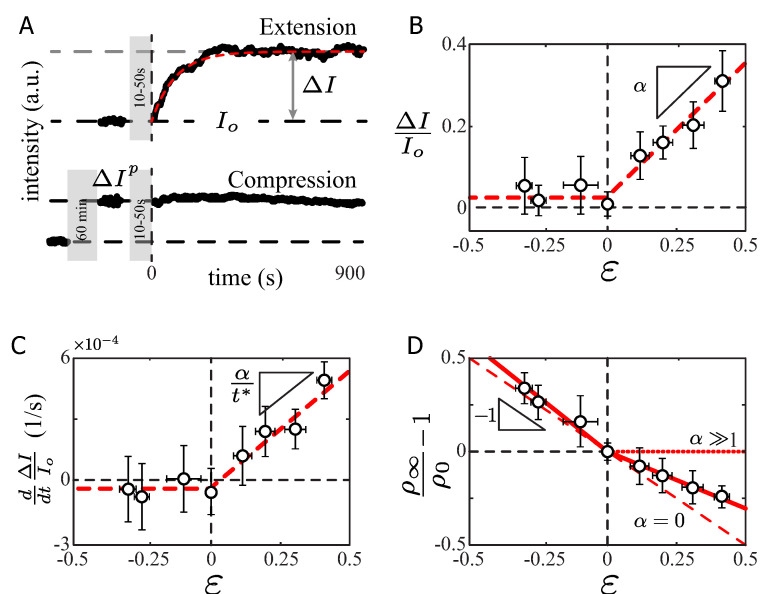
SF intensity response under stretch. (**A**) Total fluorescent intensity evolution (integrated along the bundle) at a constant stretch applied to the PDMS. (**B**) Variation of the F-actin intensity as a function of the applied strain. (**C**) Initial slope of the intensity evolution as a function of the applied strain. (**D**) Variation of the linear density, defined as the total intensity over the total SF length, a function of the applied strain. N = 12 SF for each data point.

**Figure 4 ijms-23-05095-f004:**
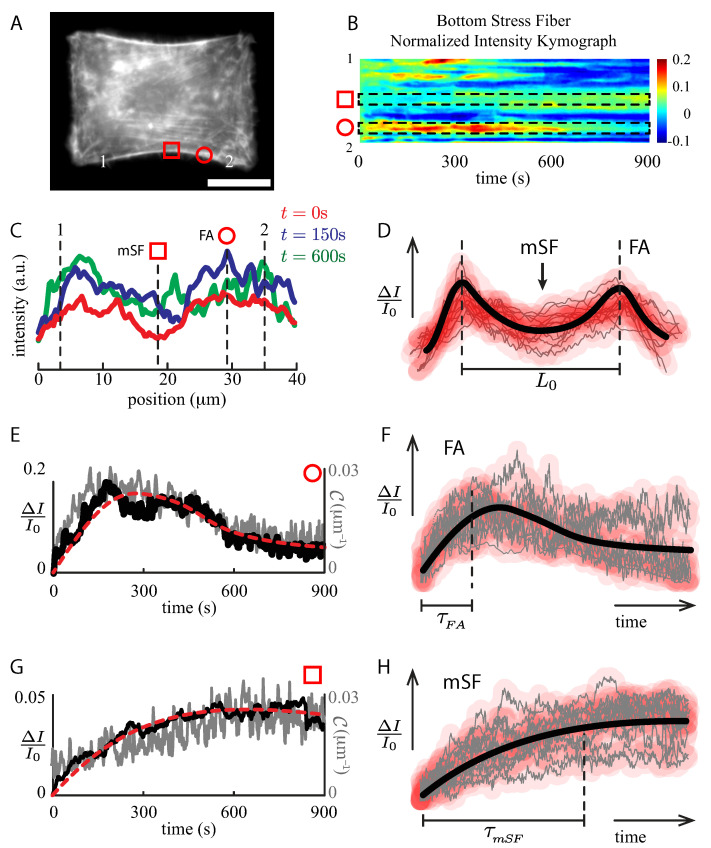
Local analysis of the F-actin intensity kymograph. (**A**) RPE-1 cell on H-like pattern at constant stretch (ϵ≈0.2). Scale bar 20μm. (**B**) Normalized relative intensity kymograph of the bottom SF (arc-segment 1–2) displays high activity at both ends of the SF, whereas a moderate intensity evolution at the middle section of the bundle. (**C**) Intensity profiles along the bottom SF for t=0 s (blue solid line), t=150 s (red solid line) and t=600 s (green solid line), respectively. The red square identifies the SF minimum intensity (mSF), whereas the red circle shows the highest F-actin intensity location (FA). (**D**) F-Actin intensity profile of 14 SFs, where the thick continuous black line represents a sketch of the F-actin bundle, with two focal adhesion sites (FAs) and a valley at the middle section of the SF (mSF). (**E**) Normalized intensity evolution of the bottom SF computed from the kymograph at the FA region (red circle), whereas (**G**) displays the evolution at the mSF region (red square). In both cases, the local curvature is shown (gray solid line). Mechanical parameter values were obtained from best fits (red segmented lines) computed using the intensity data and Equation (Equation 10). (**F**,**H**) Intensity evolution of 14 FAs and 12 mSFs locations at constant stretch. The thick continuous black line represents a sketch of the intensity evolution at these locations.

**Table 1 ijms-23-05095-t001:** Parameter values obtained from the numerical fit of Equation (Equation 10) and the experimental data using the integrated intensity of an SF segment of 5μm.

Symbol	Mechanical Parameter	Value	Model	Units
$\sigma_{0}$	Initial cortical tension	35 (SD 15)	—	pN/μm
$\lambda_{0}$	Initial line tension	1.9 (SD 1.4)	—	nN
$\nu^{-1}=\gamma /2K_{0}$	Passive time scale	320 (SD 90)	∼300	s
$\tau =\sigma_{0}R_{0}/K_{0}V$	Active time scale	240 (SD 50)	∼200	s
$f=K_{0}\epsilon_{l}/2\sigma_{0}$	Elasticity/cortex tension ratio	2–10	1	—
$\Gamma =\gamma V/2\sigma_{0}R_{0}$	Motor activity/motor dissipation ratio	0.3–1.4	1	—

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
