# Peer review of "Actin Stress Fibers Response and Adaptation under Stretch"

_ijms, 2022, doi:10.3390/ijms23095095_

Round 1
Reviewer 1 Report
This manuscript describes a nice study of SF mechanics. The topic is very interesting but more details need to be provided in order to assess this manuscript.
I feel confused about the experiment setup and more details need to be given in order for me to better review this manuscript. It would be better if the authors could provide a fig to show the experimental setup. How do the cells get stretched? Did the pulling force set to zero after stretching?
- Are the cells being stretched directly or the PDMS sheet being stretched. If the PDMS sheet is being stretched, there are multiple factors that the authors need to take into account.
- a negative control experiment must be performed to show the contribution of the PDMS to relaxation. The authors coated fibronectin on PDMS. One possible experiment is to pattern fluorescent fibronectin and track the relaxation after stretching or compressing.
- Other factors involved will be the relaxation of the cell membrane aside from the stress fiber, and also, the kinetic of bond detach/attach between the surface molecule (integrins, for example) to the ECM.
- It would be useful if the authors could provide some videos for each experiment.
- It would be helpful if the authors could include a table showing the kinetic of the SF relaxation in each condition.
- It looks like the variance in the data is pretty big. I am just wondering if the authors have assessed the variance and error associated with the model.
- I am just wondering if the authors could elaborate on how they choose to use RPE-1 cells. I haven't used this cell line myself. If this is an epithelial cell line, I am just wondering if the ideal environment for them is to form a monolayer with cell-cell junctions. If this is the case, I am wondering if the single-cell patterned culture will affect the cell behaviors. I would suggest the authors test more cell types.
Author Response
- Are the cells being stretched directly or the PDMS sheet being stretched. If the PDMS sheet is being stretched, there are multiple factors that the authors need to take into account.
- a negative control experiment must be performed to show the contribution of the PDMS to relaxation. The authors coated fibronectin on PDMS. One possible experiment is to pattern fluorescent fibronectin and track the relaxation after stretching or compressing.
In our experiments, the PDMS sheet (30mmX20mmX.15mm) is submitted to an initial stretching ratio of 100um/s, then the stretching stops and the elongation (or contraction) is maintained by 15 minutes at a temperature of 37ºC. In our conditions, the PDMS residual strain of a creep test is negligible. In deed, residual strain is noticeable over temperatures of at least 80ºC after 30 minutes of continuous stretching (https://www.nature.com/articles/s41598-017-11485-6)
This reference is now included in the paper.
We have included a comment at PAGE 2, Line 55
- Other factors involved will be the relaxation of the cell membrane aside from the stress fiber, and also, the kinetic of bond detach/attach between the surface molecule (integrins, for example) to the ECM.
In the range of explored stretches and experiment duration, we do not observe relaxation of the length due to a detachment of the cell from the micropatterned PDMS substrate.
We have included a comment at PAGE 2, Line 59
2. It would be useful if the authors could provide some videos for each experiment.
We now include two videos of extreme situations: an experiment for 40% stretch and another experiment for 30% of contraction.
We have included a reference to the video at PAGE 2, Line 55 and at PAGE 4, Line 130
3. It would be helpful if the authors could include a table showing the kinetic of the SF relaxation in each condition.
We have included the table at PAGE 8. Also, a comment at PAGE 9, Line 337
4. It looks like the variance in the data is pretty big. I am just wondering if the authors have assessed the variance and error associated with the model.
The parameters are measured or inferred directly from the Intensity over time or Radius of curvature over time data. Thus, the variance between experiments corresponds to the natural variance of stress fibers samples. As mentioned in the figures, the error bars stand for the Standard Variations. The uncertainty is the Standard Deviation divided by the square root of the number of Stress Fibers experiments. In other words, the error bars should be divided by a factor equal to √(12)≅3.5. However, if we do this, the error bar will not be noticeable.
We have included a clarification in the description of figure 2.
5. I am just wondering if the authors could elaborate on how they choose to use RPE-1 cells. I haven't used this cell line myself. If this is an epithelial cell line, I am just wondering if the ideal environment for them is to form a monolayer with cell-cell junctions. If this is the case, I am wondering if the single-cell patterned culture will affect the cell behaviors. I would suggest the authors test more cell types.
Human retinal pigment epithelial-1 (RPE-1) cells are being used as a model to study mitosis , cytoskeleton organization (https://www.nature.com/articles/ncb1307 , https://www.nature.com/articles/ncb2269, https://www.pnas.org/doi/10.1073/pnas.0609267103 ). Epithelial as cells highly spread one ECM on the retina but they doesn’t form tight junction and have weak cell-cell interaction. They represent a non-transformed alternative to cancer cell lines, such as HeLa cells.
We include these 3 references in the paper.
We have included a comment at PAGE 2, Line 55
Reviewer 2 Report
In this paper, Roberto Bernal and colleagues investigated how actin stress fibers in mammalian cells respond to the external stretch. The authors used fluorescence analysis and a theoretical model to explain the response mechanism of focal adhesion. They concluded the response is highly dependent on the type of mechanical stress. In addition, they also proposed that the repair of the actin bundle is happening at the weak point of stress fibers and the formation of a new actin bundle will lead to the strain-softening on stress fibers. The result in this paper is convincing and well presented. The model in this work is plausible. Accordingly, I recommend publication of this manuscript after addressing the very few concerns below:
Major issues:
(1) To better ensure the conclusion is universally true, apart from the RPE-1 cells, I would suggest the authors also conduct the same analysis on several more cell types.
(2) Do the authors expect the mechanical property of the PDMS pad affects the cell behaviors? Will changing physical parameters of the PDMS (e.g. Young’s modulus) also alter the conclusion of this paper?
(3) The authors might consider including some drug experiments to better support the conclusion in this paper. For example, after the formation of mature focal adhesion, if an inhibitor such as CK666 (Arp2/3 inhibitor) is introduced, the actin branching will generally be halted. This will presumably shut down the incorporation of new actin into the stress fibers. In this case, will the authors observe the reduction of strain-softening?
Minor issues:
(4) Page 9, line 368. Typo: “At the middle section of the atin bundle (mSF)…” the “atin” should be “actin”.
Author Response
1. To better ensure the conclusion is universally true, apart from the RPE-1 cells, I would suggest the authors also conduct the same analysis on several more cell types.
Human retinal pigment epithelial-1 (RPE-1) cells are being used as a model to study mitosis, cytoskeleton organization (https://www.nature.com/articles/ncb1307 , https://www.nature.com/articles/ncb2269, https://www.pnas.org/doi/10.1073/pnas.0609267103 ). Epithelial as cells highly spread one ECM on the retina but they don’t form tight junction and have weak cell-cell interaction. They represent a non-transformed alternative to cancer cell lines, such as HeLa cervical adenocarcinoma cells. We have included these references into the paper.
We have included a comment at PAGE 2, Line 55
2. Do the authors expect the mechanical property of the PDMS pad affects the cell behaviors? Will changing physical parameters of the PDMS (e.g. Young’s modulus) also alter the conclusion of this paper?
There are indeed several papers showing that the cells morphology depends on the Young Modulus, stress field and substrate chemistry, where the cell-substrate interaction is non-specific or in cases that the substrate elasticity resembles the properties of soft tissue (https://www.nature.com/articles/s41467-018-02906-9). However, using Théry’s approach for cell surface interaction, isolated cell interacting with a specific bond (fibronectin) diminishes the chances to, for example, cell differentiation due to the substrate elasticity observed in stem cells (https://www.nature.com/articles/ncb1307). Moreover, soft PDMS displays elastic properties that are 10-fold over than soft tissue (Brain Tissue E~1KPa, Hidrogels E<10KPa, soft PDMS E>1MPa), with non or very little viscoelastic effects for PDMS at room temperature in a creep test (https://www.nature.com/articles/s41598-017-11485-6). Therefore, we should not expect a significant dependence on the mechanical properties of the substrate in our model.
These references are now part of the paper
We have included a comment at PAGE 2, Line 55
3. The authors might consider including some drug experiments to better support the conclusion in this paper. For example, after the formation of mature focal adhesion, if an inhibitor such as CK666 (Arp2/3 inhibitor) is introduced, the actin branching will generally be halted. This will presumably shut down the incorporation of new actin into the stress fibers. In this case, will the authors observe the reduction of strain-softening? 

We performed drug experiments however they showed a high variability (except for the use of phalloidin as a positive control increasing the intensity variation over 30% at a constant stretch value of ε=0.1 ), thus we preferred not to present the data. We should reperform and continue this side of the study, however the person in charge of this part moved out the lab.
Minor issues:
4. Page 9, line 368. Typo: “At the middle section of the atin bundle (mSF)…” the “atin” should be “actin”.
Corrected
Reviewer 3 Report
The authors describe “actin stress fibers response and adaptation under stretch” using RPE-1 cells.
This is an interesting study in an area that needs investigating. This is also a carefully written manuscript, and the findings are of considerable interest.
A few minor revisions are listed below.
Figure 1 and Figure 2:
For the reader’s convenience image size of Figures 1A and 1B should be enlarged to show stress fibers more clearly. We can not see stress fibers in Figures 1 A and B. Focal adhesion should be indicated in Figures using anti-paxillin or anti-vinculin for the marker of focal adhesions if the authors discuss FAs in this figure. It is also helpful if the author indicates typical stress fibers in Figure 1 A and B. Stress fibers in Figure 2A are also unclear because the magnification of fluorescent images is small.
Figure 4 is a very exciting results.
In the Materials and Methods section, a description of how to stretch mechanically PDMS sheets is lacking. Detailed methods to stretch substrate should be included in the manuscript. Images of the stretching device are also helpful to understand the stretch to the substrate.
Author Response
1. For the reader’s convenience image size of Figures 1A and 1B should be enlarged to show stress fibers more clearly. We can not see stress fibers in Figures 1 A and B. Focal adhesion should be indicated in Figures using anti-paxillin or anti-vinculin for the marker of focal adhesions if the authors discuss FAs in this figure.
We should perform labeling experiment, however the person in charge of this part moved out the lab.
We have modified the Figure 1 including a green circle to indicate the location of the FAs. In figure 2, we have shift the position of the segmented red lines to allow the visualization of the SF at the free edge of the cells. We have also included a comment in the caption of figure 1 indicating the FAs location.
2. It is also helpful if the author indicates typical stress fibers in Figure 1 A and B. Stress fibers in Figure 2A are also unclear because the magnification of fluorescent images is small.
We have enhanced the contrast the images and increased the size of figure 2 to help the reader.
3. Figure 4 is a very exciting results.
Thanks!
4. In the Materials and Methods section, a description of how to stretch mechanically PDMS sheets is lacking. Detailed methods to stretch substrate should be included in the manuscript. Images of the stretching device are also helpful to understand the stretch to the substrate.
We use the same device as in the following papers:
Fink et al. 2011 https://www.nature.com/articles/ncb2269, Azioune et al. 2011 https://pubs.acs.org/doi/abs/10.1021/la200970t, Dejadin et al. 2020 https://rupress.org/jcb/article/219/10/e201908036/152020/
These references are now incorporated into the text in PAGE 10, Line 412
Round 2
Reviewer 1 Report
The authors have addressed most of my concerns. Here are some follow-up comments.
2. Please add timestamp and scale bar to the videos.
4. Please include details about how to subtract the effect of photobleaching in the method section. Also, please list n=x (number of SFs) and N=y(number of cells). In my opinion, at least 10 cells are needed.
5. Both reviewers have raised the comment on using different cell types. I am ok that the authors don't want to put extra effort to include more cell types. Please include a discussion to talk about the potential limitations (cell type-specific).
Author Response
2. Please add timestamp and scale bar to the videos.
The supplementary videos now present the scale bar and timestamp
4. Please include details about how to subtract the effect of photobleaching in the method section. Also, please list n=x (number of SFs) and N=y(number of cells). In my opinion, at least 10 cells are needed.
A reference has been included with the details of the method used to correct the bleaching (https://f1000research.com/articles/9-1494/v1).
Page 10, Line 435
Page 11, Line 438. Unless otherwise mentioned, the number of Stress Fibers is equal to the number of cells.
5. Please include a discussion to talk about the potential limitations (cell type-specific).
Two paragraphs have been included relating to the RPE-1 cell line
Page 2, Line 55. RPE-1 cell line to study mechanically, biochemically, and the reorganization of the actin cytoskeleton.
Page 8, Line 311. Limitations of our experimental approach. Two new references have been included on how to approach wound healing and non-peripherical stress fibers (10.1016/j.bpj.2013.11.015, 10.1126/science.abb2169).